# Syntactic denoising and multi-strategy auxiliary enhancement for aspect-based sentiment analysis

**Lu Liu**[1], **Da Li**[1], **Chuanxu Yue**[1], **Xiaojin Gao**[2], **Yunhai Zhu**[2]*

**1** Institute of Automation, Qilu University of Technology (Shandong Academy of Sciences), JiNan, ShanDong, China, **2** Science and Technology Service Platform, Qilu University of Technology (Shandong Academy of Sciences), JiNan, ShanDong, China

* zhuyh@sdas.org

**Data availability statement:** The datasets used in this paper can be accessed at the following address: https://github.com/doctortai/SDMAE.git.

## Abstract

Aspect-based sentiment analysis (ABSA) aims to identify the sentiment polarity associated with specific aspect terms within sentences. Existing studies have primarily focused on constructing graphs from dependency trees of sentences to extract syntactic features. However, given that public datasets are often derived from online reviews, the syntactic structures of these sentences frequently exhibit irregularities. As a result, the performance of syntactic-based Graph Convolution Network (GCN) models is adversely impacted by the noise introduced during dependency parsing. Moreover, the interaction between syntactic and semantic information in these approaches is often insufficient, which significantly impairs the model's ability to accurately detect sentiment.To address these challenges, we propose a novel approach called Syntactic Denoising with Multi-strategy Auxiliary Enhancement (SDMAE) for the ABSA task. Specifically, we prune the original dependency tree by focusing on context words with specific part-of-speech features that are critical for conveying the sentiment of aspect terms, and then construct the graph. We introduce a Multi-channel Adaptive Aggregation Module (MAAM), a feature aggregation system that employs a multi-head attention mechanism to integrate semantic and syntactic GCN output representations. Furthermore, we design a multi-strategy task learning framework that incorporates sentiment lexicons and supervised contrastive learning to enhance the model's performance in aspect sentiment recognition.Comprehensive experiments conducted on four benchmark datasets demonstrate that our approach achieves significant performance improvements compared to several state-of-the-art methods across all evaluated datasets.

## Introduction

Sentiment analysis, as a pivotal research area in Natural Language Processing (NLP), has garnered considerable attention from scholars both domestically and internationally. The

**Funding:** The Key R&D Program (Science and Technology Cooperation) of Shandong Province (2024KJHZ030). The Innovation Pilot Project for the Integration of Science, Education, and Industry (2024GH12). The Innovation Capability Enhancement Project for Science and Technology oriented Small and Medium sized Enterprises in Shandong Province (2024TSGC0903).

**Competing interests:** The authors have declared that no competing interests exist.

rapid proliferation of the internet has led to the emergence of various platforms, such as e-commerce [1,2], social forums [3–5], and online healthcare [6,7]. As a result, online comments have surged dramatically, often containing subjective evaluations of products by consumers or opinions on major social events and government decisions. Consequently, there is an increasing demand within the industry to extract intricate sentiment patterns from this data. Such capabilities would enable management in various sectors, including e-commerce, social forums, and governmental organizations, to enhance and adjust their products and policies accordingly.

Traditional sentiment analysis tasks primarily focus on discerning coarse-grained sentiment polarity at the sentence level. However, these tasks face considerable challenges when addressing the complexity and variability of online reviews, which has spurred the advancement of the Aspect-based Sentiment Analysis (ABSA) task. ABSA aims to determine the sentiment polarity associated with specific aspect terms within a sentence. For instance, as illustrated in Fig 1.

The sentence "Apple's products look great, except that the charging speed is a little slow" contains two aspect terms: "products," which is associated with a positive sentiment polarity, and "speed," which is linked to a negative sentiment polarity. Traditional sentiment analysis techniques struggle with sentences that express multiple, conflicting sentiments. In contrast, the ABSA task requires models to identify the sentiment orientations associated with particular entities in online consumer reviews, rather than delivering a generalized sentiment assessment of the entire sentence.

Initial methodologies for ABSA were labor-intensive, including lexicon-based, rule-based, and machine learning-based approaches. The effectiveness of these techniques heavily relied on the quality of feature engineering, which limited their generalization and transferability. The advent of deep learning has led to the development of sequence-based neural network models, such as Recurrent Neural Networks (RNNs) and their variants, such as Long Short-Term Memory (LSTM) networks. These models enable the automated extraction of semantic information from textual data. Researchers have increasingly employed these models to autonomously extract semantic information related to both context and aspect terms. To facilitate the automatic identification of key word features within textual data, attention-based approaches have been extensively explored. The development of various attention mechanisms specifically tailored for ABSA has led to significant improvements in its performance.

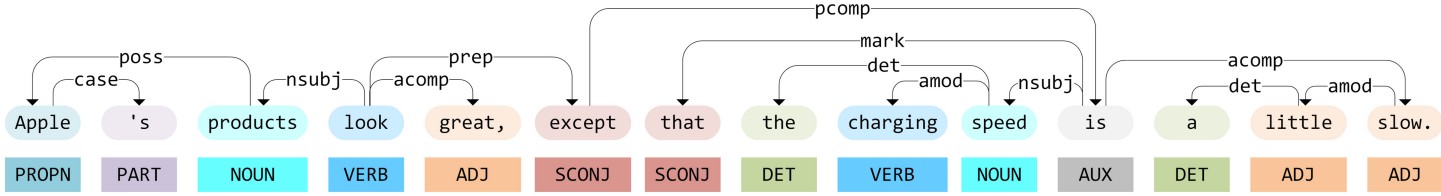

**Fig 1. The demonstration of ABSA.** The sentence "Apple's products look great, except that the charging speed is a little slow." has two aspects: products and speed, with their sentiment polarities being positive and negative, respectively.

Inspired by feature extraction techniques in computer vision, researchers have begun applying Convolutional Neural Networks (CNNs) to the ABSA task. This approach involves extracting various local features from textual data using sliding convolutional kernel windows. In 2017, Zhang et al. [8] developed an undirected graph derived from the dependency tree of sentences and were the first to apply Graph Convolutional Networks (GCNs) to model contextual dependencies, yielding notable results. This pioneering work has driven a significant increase in the use of GCNs by researchers in recent years to capture syntactic information in ABSA.Recent studies have also integrated external knowledge bases, including sentiment lexicons and knowledge graphs, into ABSA models to incorporate commonsense knowledge. These approaches [9–15] adjust weights within the syntactic dependency graph or establish connections between entities, enabling models to prioritize significant opinion words and their relationships with entities.

It is important to note that, although these methods have demonstrated notable enhancements in performance, several challenges persist that have yet to be adequately addressed:

1. There is a considerable presence of noisy information within the dependency trees of informal online reviews.
2. The significance of opinion words with sentiment tendencies is frequently neglected in the analysis of semantic information.
3. The interplay between semantic and syntactic information in the text is not sufficiently explored.

To address the challenges in the ABSA task outlined above, we propose SDMAE, which integrates syntactic denoising with a multi-strategy auxiliary enhancement. Specifically, to mitigate noisy information within dependency trees derived from online reviews, we introduce a syntactic denoising pruning strategy. This methodology eliminates extraneous word dependencies by incorporating part-of-speech and positional distance attributes, thereby constructing a syntactic denoising graph.Subsequently, we employ a pre-trained BERT model to encode the text and create a semantic graph through multi-head attention operations applied to the textual semantic features. Graph Convolutional Networks (GCN) are then utilized to extract features from both the semantic graph and the syntactic denoising graph. The semantic and syntactic features are subsequently integrated via the Multi-channel Adaptive Aggregation Module (MAAM).To effectively capture sentiment information conveyed by opinion words in context, we propose a sentiment refinement strategy. Using the SenticNet 8 sentiment lexicon, we generate a sentiment vector for each text in the dataset and compute the mean squared error loss between this vector and the output produced by the adaptive aggregation module. This strategy enhances the model's ability to identify and incorporate key sentiment indicators. Additionally, to improve the model's capacity to distinguish between different sample classes, we incorporate supervised contrastive learning as an auxiliary strategy during training.

To summarize, the main contributions of our work are as follows:

- We introduce SDMAE, an innovative approach to ABSA that integrates syntactic denoising with a multi-strategy auxiliary enhancement framework. This method markedly improves the performance of ABSA.
- We propose an aspect-oriented syntactic denoising algorithm that efficiently eliminates extraneous noise information from the dependency tree. This is achieved by integrating words with their corresponding parts of speech and their non-linear positional attributes about aspect words, thereby facilitating the pruning of the dependency tree.

- We design an affective refinement strategy module that employs a sentiment lexicon to aid the model in recognizing significant sentiment indicators within the contextual framework. The integration of this module with supervised contrastive learning techniques enhances the overall training process of the model.
- Extensive experiments on REST14, Twitter, REST15, and REST16 four benchmark datasets demonstrate the effectiveness of our proposed methodology in the ABSA task.

## Related works

Sentiment analysis can be classified into two categories based on the granularity of sentiment entities: coarse-grained and fine-grained analysis. Coarse-grained sentiment analysis includes both document-level and sentence-level sentiment evaluations. In contrast, ABSA primarily focuses on sentiment entities at the word level, which presents greater challenges compared to document-level and sentence-level sentiment analysis.As ABSA research gains momentum, traditional approaches that rely on manual feature construction are increasingly being phased out due to their limited generalization across domains and the substantial labor costs involved. The advent of deep learning has paved the way for new methodologies and approaches in the ABSA task. In this section, we provide an overview of the prevailing deep learning-based methodologies for ABSA, including neural network-based approaches, GCN-based models, and methods that incorporate external knowledge.

### Neural network-based methods

Sequential models are commonly employed for analyzing sequential data, such as text and speech. Unlike conventional methods that rely on manually designed features, neural network-based approaches demonstrate strong generalization abilities and have achieved remarkable performance. For instance, Tang et al. [16] utilize two LSTM networks to capture information from both sides of the aspect, combining this information to form the final representation for sentiment classification. However, methods relying solely on sequence models struggle to capture the importance of contextual word information.Since attention mechanisms can prioritize critical components of sentences, researchers have explored integrating sequence models with attention mechanisms. Wang et al. [17] combine aspect embeddings with sentence representations, merging the attention mechanism with LSTM networks to extract significant contextual semantic features. Huang et al. [18] introduce the concept of attention over-attention, derived from machine translation, to simultaneously model context and aspect features using LSTM networks. Ma et al. [19] leverage LSTM to obtain representations of both context and aspects, employing an interactive attention network to extract contextual information that significantly influences aspect sentiment polarity. Chen et al. [20] use Bidirectional Long Short-Term Memory (BiLSTM) networks to model contextual information and implement multiple attention mechanisms to effectively capture long-range context dependencies. Fan et al. [21] design a multi-granularity attention mechanism that combines coarse-grained and fine-grained attention to enhance model performance. Zhu et al. [22] propose a CNN for phrase extraction and introduce a cross-correlation attention mechanism, which allocates weights to phrases based on words in the context and adjusts weights for individual words in the context according to the phrases.The integration of attention mechanisms has resulted in significant improvements in ABSA task performance, enhancing the model's ability to capture relevant semantic features from both context and aspect terms.

## GCN-based methods

Despite notable advancements in performance from the integration of attention mechanisms, approaches focusing solely on semantic aspects continue to face challenges in effectively capturing long-distance syntactic dependencies within the context. Graph Convolutional Networks (GCN) have been widely adopted for the ABSA task due to their ability to model long-distance dependencies in the dependency tree. Zhang et al. [8] convert the dependency tree into an adjacency graph and apply GCN to model this graph, yielding excellent results. To provide the model with additional contextual information, Zhang et al. [23] propose incorporating conceptual hierarchies of syntax and lexicon, thereby constructing hierarchical syntactic and lexical graphs. They then develop a bi-level GCN aimed at the comprehensive integration of information from both the hierarchical syntactic and lexical graphs. Pang et al. [24] introduce a dynamic multi-channel GCN model that separately models syntactic and semantic information, and design a parameter-sharing GCN to extract common information, which is then concatenated after average pooling for ABSA. Li et al. [25] propose a dual-channel GCN to model semantic and syntactic graphs, employing orthogonal and differential regularizers to aid the model in thoroughly learning semantic features.

Since both semantic and syntactic features can be influenced by irrelevant words, edge effects, and other local factors, Wang et al. [26] introduce a distance-based syntactic weighting algorithm to prune the dependency parse tree. By combining aspect-fusion attention, they further filter opinion words in the context, achieving precise identification of aspect terms. In the ABSA task, the sentiment polarity associated with aspect terms is sometimes contingent upon specific contextual phrases. To avoid introducing irrelevant context and syntactic dependencies, researchers have focused on extracting localized segment information relevant to specific aspects. Ahmad et al. [27] propose a specific aspect-based segmentation framework that segments the sentence, retaining only the portion related to the particular aspect, and then uses GCN to extract both syntactic and semantic features.

You et al. [28] demonstrate that dependency trees can introduce extraneous noise due to irrelevant associations, which may lead to erroneous alignments between aspects and their associated sentiment words. To address this, they employ sentiment-aware contextual trees, incorporating phrase segmentation and hierarchical structures alongside graph attention mechanisms. This approach allows the model to effectively capture detailed syntactic information from both contextual and dependency trees, thereby ensuring precise alignment between aspect terms and their respective sentiment words.

Despite the notable performance improvements achieved by graph-based methods in the ABSA task, challenges remain in effectively leveraging external knowledge to enhance the models' capabilities in sentiment recognition.

## External knowledge-based methods

External knowledge has been shown to enhance the natural language understanding abilities of models, with wide-ranging applications across various NLP tasks, such as event detection [29], text classification [30], and more. To optimize the utilization of external resources, including sentiment lexicons [31,32] and knowledge graphs [33,34], researchers have proposed a variety of strategies to improve the integration of external knowledge into neural networks in a more comprehensive and efficient manner. Ma et al. [10] integrate the LSTM network with the SenticNet 4 [35] sentiment lexicon by enhancing LSTM units to incorporate sentiment information into deep neural networks. Following this, approaches utilizing graph neural networks (GNNs) have become the dominant strategy for tackling the ABSA task, prompting researchers to explore the potential of combining GNNs with external knowledge

sources. Zhou et al. [36] combine a syntactic dependency graph with commonsense knowledge graphs using GCN. Zhong et al. [15] propose a multi-view representation enhancement network that integrates knowledge graphs into the embedding space, further facilitating the acquisition of aspect-specific knowledge representations through attention mechanisms. Liang et al. [11] employ the SenticNet 6 [37] sentiment lexicon to adjust the weights within the dependency parse graph. Their experimental findings show that incorporating sentiment information helps the model prioritize opinion words with significant sentiment relevance. Gu et al. [12] integrate sentiment knowledge and part-of-speech information into the original syntactic dependency graph, resulting in an augmented graph that incorporates syntactic, sentiment, and part-of-speech information simultaneously. Zheng et al. [38] develop a framework for incorporating sentiment knowledge at the corpus level by utilizing sentiment lexicons [32]. This framework enhances the model's ability to retain, modify, and share sentiment knowledge, enabling the transfer of sentiment knowledge during training on different datasets within the same domain by initializing new model nodes with pre-existing sentiment knowledge node representations. Hao et al. [13] propose a network model that integrates three channels of GCN, leveraging the Concept knowledge graph [34] alongside the SenticNet sentiment lexicon. This model effectively captures contextual semantic features, conceptual knowledge, and sentiment knowledge, thereby enhancing its ability to express aspects and represent sentence dependency graphs. Additionally, the incorporation of interactive attention mechanisms further optimizes the coordination between aspects and context.

The aforementioned studies suggest that a model's sentiment analysis capabilities can be significantly improved through the integration of external knowledge. However, there is still a need for further exploration of effective knowledge fusion methods.

Existing methods merely assign contextual position encoding and sentiment lexicon scores to an adjacency matrix in a simplistic manner, overlooking contextual noise information and contrastive learning of sentiment knowledge. Therefore, our SDMAE model employs Gaussian functions to mitigate noise information, while leveraging contrastive learning to fuse sentiment knowledge, considering the utilization of both for richer feature information.

## Methodology

Task Description: Given a sentence $S$ and aspect term $A$, where $S$ denotes the sentence of length $n$ and contains $m$ aspect terms. $S = \{w_1, w_2, w_3, ..., w_{t+1}, ..., w_{t+m}, ..., w_n\}$, where $A = \{w_{t+1}, ..., w_{t+m}\}$ is the subsequence of $S$. The goal of the ABSA task is to identify the sentiment polarity of each aspect term $w_i \in A$ in the sentence. Fig 2 illustrates the framework of our proposed method SDMAE, which includes five key components: 1) **Encoder Layer**: This layer encodes the input sentence into a suitable representation for further processing. 2) **Dual-channel GCN Layer**: This layer employs two parallel GCNs to model semantic and syntactic graphs separately. 3) **Multi-channel Adaptive Aggregation Module**: This module aggregates features from both semantic and syntactic GCN adaptively. 4) **Output Layer**: This layer generates the final sentiment prediction representation based on the aggregated features. And 5) **Multi-Strategy Auxiliary Module**: This module incorporates additional strategies to enhance the model's performance. Next, we will provide detailed descriptions of each component of SDMAE.

### Encoder layer

For a given sentence encoded using pre-trained BERT [39], we follow BERT-SPC [40] approach to pre-process the text, concatenate the sentence with aspect words, and encode the

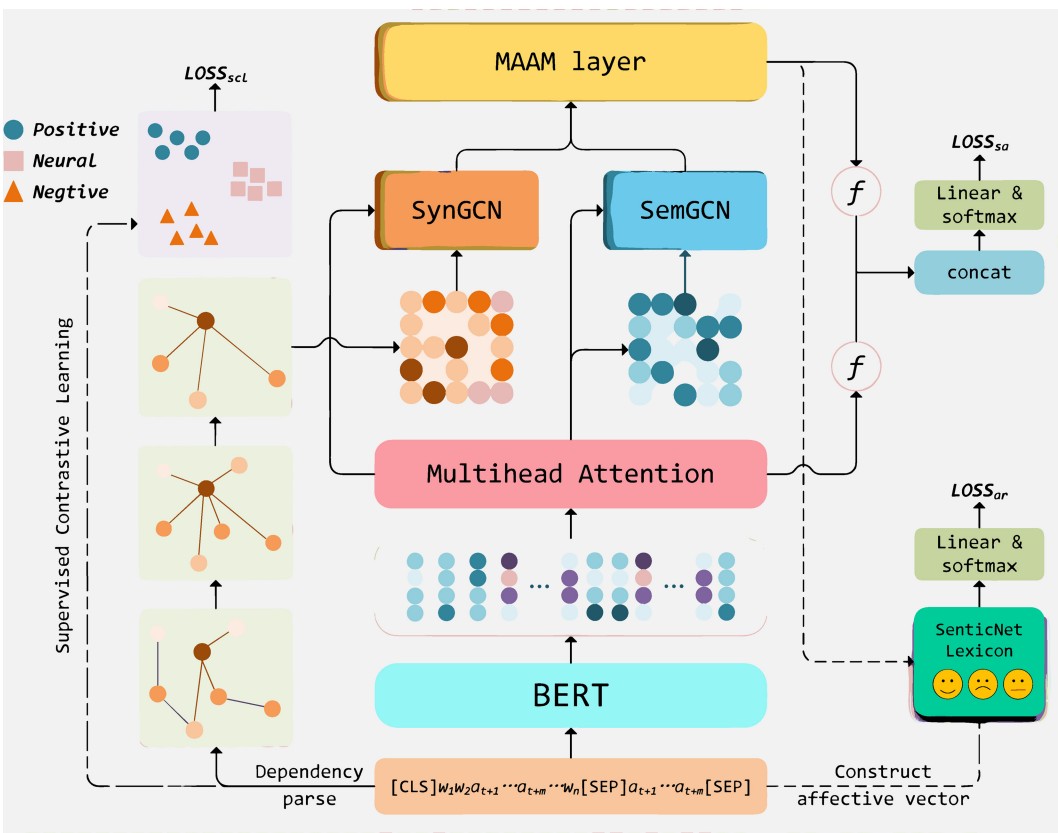

**Fig 2. The overall structure of SDMAE.** In sentence-aspect pairs encoder, mainly a BERT-based encoding process. DualGCN modules contain SemGCN and SynGCN. $Loss_{ar}$ and $Loss_{scl}$ refer to the multi-strategy auxiliary enhancement loss.

concatenated representation.

$$input_{BERT} = [CLS] + Sentence + [SEP] + Aspect + [SEP] \quad (1)$$

where $[CLS]$ and $[SEP]$ are the special tokens in the pre-trained BERT, which denote the sentence-pooling representation of the fine-tuned BERT for the downstream classification task as well as the tokens used for separating the two sentences, respectively. The last hidden layer of BERT states $H$ is obtained as the semantic feature representation of sentence-aspect pair: $H = \{h_1, h_2, h_3, ..., h_{t+1}, ..., h_{t+m}, ..., h_n\}$, where $h_0$ denotes the output of pooling i.e. the vector representation at $[CLS]$. The word vectors obtained after BERT has more powerful representations compared to GloVe word vectors and can automatically learn the semantic, positional, part-of-speech, and syntactic information of the text.

Given that linear encoding offers a restricted amount of learnable information throughout the model training process, we employ a Gaussian function-based positional weighting, which is calculated as demonstrated below:

$$distance_i = \begin{cases} |1-(1+t-i)|/_n, & 1 \leq i < t+1 \\ 0, & t+1 \leq i \leq t+m \\ |1-(i-t-m)|/_n, & t+m < i \leq n \end{cases} \quad (2)$$

$$p(\cdot) = \exp(-distance^2/(2\sigma^2)) \tag{3}$$

following the application of the function $p(\cdot)$, we derive semantic information with nonlinear distance features, which demonstrates greater efficacy compared to linear feature perception and enables the modulation of weight distribution through the adaptive adjustment of the $\sigma$ parameter.

## Dual-channel GCN layer

**SemGCN.** To improve the representation of semantic features and aid the model's understanding, we develop SemGCN using a multi-head attention mechanism. We create a semantic graph by assessing the scores from various attention heads, where the attention scores between each word pair reflect their semantic relevance, the calculation process is outlined as follows:

$$H^{att}, score = \text{MHA}(h_i) \tag{4}$$

$$score_i = \text{softmax}\left( \frac{Q_i W_i^Q \times (K_i W_i^K)^T}{\sqrt{d_k}} \right) \tag{5}$$

$$H^{att} = score_i V_i \tag{6}$$

$$A^{sem} = \frac{\sum_{i=1}^{heads} score_i}{heads} \tag{7}$$

where $Q$, $K$, and $V$ are derived from the semantic features $H$ through linear transformations. Then put them into the multi-head attention mechanism, yielding the multi-head attention output denoted as $H^{att}$. By averaging the attention scores $score_i$ from each attention head, we construct the semantic graph $A^{sem}$ of the sentence. To acquire semantic graph information, we employ GCN for modeling the semantic graph, denoted as SemGCN. The position function $p(\cdot)$ is utilized to process the multi-head attention output $H^{att}$ and derive the initial representation $\widetilde{H}$ for SemGCN. The hidden states of the SemGCN are updated as follows:

$$\widetilde{H} = p(H^{att}) \tag{8}$$

$$\widetilde{h_i^{sem}} = \text{ReLU}(A_i^{sem} g_i^{l-1} W^l + b^l) \tag{9}$$

where $W^l$ and $b^l$ represent the learnable weight and bias matrices, respectively. $A_i^{sem} = A^{sem}/E_i + 1$, $E_i$ is the degree matrix of the semantic graph $A^{sem}$, and $g_i^{l-1}$ represents the output of node $i$ from layer $l$-1. Upon traversing through $l$ layers of SemGCN, the final output of SemGCN is derived, $\widetilde{H^{sem}} = \{\widetilde{h_1^{sem}}, \widetilde{h_2^{sem}}, \widetilde{h_3^{sem}}, \cdots, \widetilde{h_n^{sem}}\}$.

**SynGCN.** The dependency tree provides syntactic dependency information that illustrates the relationships between words within a sentence, thereby enabling the connection of two distant words through specific types of dependency relations. In the context of the ABSA task, the majority of dependency parse trees for sentences are generated using NLP toolkits, such as spaCy or Stanford NLP, which are then employed to construct syntactic graphs. The general rule is that if there is a dependency arc connecting word $i$ and word $j$ in the parse tree, the corresponding position $A_{i,j}$ in the adjacency matrix $A$ is set to 1; otherwise, it is set to 0. The construction of $A_{i,j}$ can be computed as follows:

$$A_{i,j} = \begin{cases} 1, & \text{self-loop} \\ 1, & \text{if } (w_i, w_j) \in \text{Rel} \\ 0, & \text{otherwise} \end{cases} \tag{10}$$

where "self-loop" represents that the position $A_{i,i}$ corresponding to word $i$ in the matrix is 1, and "Rel" represents specific type of dependency relation between the words.

However, it is noteworthy that the experimental dataset derived from online comments encompasses a substantial amount of noisy information. This includes numerous words that are unrelated to aspect sentiment, dependencies, and non-standard syntactic structures. Consequently, denoising the original dependency tree is imperative. We propose an Aspect-Oriented Syntactic Denoising (AOSD) algorithm that integrates aspect information, part-of-speech tags, and distance weights to reconstruct and prune the dependency trees.

Algorithm 1 delineates the detailed procedure of our proposed AOSD method. We employ spaCy to generate the dependency tree $T$ from the original sentence and predefine a set of part-of-speech tags, concentrating on specific word information. The selection of part-of-speech tags is based on research conducted by Gu and Shuang et al. [41,42], which demonstrated that words belonging to part-of-speech tags such as adjectives, adverbs, and verbs significantly influence the sentiment orientation toward a given aspect.

**Algorithm 1. Aspect-oriented syntactic denoising.**

1: **Input:** sentence $S = \{w_1, w_2, \dots, w_n\}$, aspect $A = \{w_{t+1}, \dots, w_{t+m}\}$, dependency Tree $T$ (parsed by spaCy), *Part-of-Speech_List* = [VERB, ADV, ADJ, ADP]

2: **Output:** Aspect-Oriented Syntactic Denoising matrix $M$

3: Initialize a zero matrix $M \in \mathbb{R}^{n \times n}$

4: **for** $i = 1$ to $n$ **do**

5: **for** $j = 1$ to $m$ **do**

6: direct $w_i.head = w_j$ in $T$

7: **if** $w_i.pos\_tag$ not in Part-of-Speech_List **then**

8: $score \leftarrow 0$

9: **else**

10: $score \leftarrow 1$

11: **end if**

12: $M_{i,j} \leftarrow \frac{|i-j|}{n} + score$

13: **end for**

14: $M_{i,i} \leftarrow 1$

15: **end for**

16: **return** $M$

Firstly, we traverse the words within the sentence to reconstruct the dependency tree, establishing a linkage between each node and the aspect terms. Subsequently, we eliminate the dependencies of nodes that do not fall into the predefined part-of-speech list. Eventually, we create the dependency matrix $M$, in which the weight between nodes and aspect terms is computed based on the relative distance between the nodes and the aspects. The visualization of the AOSD algorithm is depicted in Fig 3.

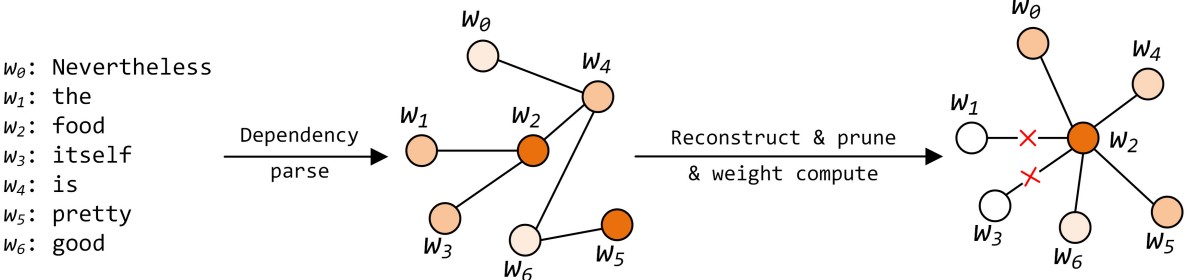

**Fig 3. Visualization process of AOSD algorithm.**

By employing Algorithm 1, we construct a syntactic denoising adjacency matrix M. Analogous to SemGCN, we also utilized $\widetilde{H}$ as the initial input for SynGCN. The node update equation of SynGCN is presented by the following equation:

$$\widetilde{h_i^{syn}} = \text{ReLU}(M_i^{syn} g_i^{l-1} W^l + b^l) \tag{11}$$

where $W^l$ and $b^l$ represent the learnable weight and bias matrices, respectively. And $g_i^{l-1}$ represents the output of node $i$ from layer $l$-1. After traversing $l$ layers of SynGCN, the final output of SynGCN is derived. $\widehat{H^{syn}} = \{\widetilde{h_1^{syn}}, \widetilde{h_2^{syn}}, \widetilde{h_3^{syn}}, \cdots, \widetilde{h_n^{syn}}\}$.

## MAAM layer

In previous dual-channel ABSA approaches, the aggregation of semantic and syntactic modules was predominantly accomplished via simple concat or summation, or through the design of gate mechanisms. Nevertheless, these methods have failed to achieve satisfactory performance, signifying their limitations in facilitating mutual learning between semantic and syntactic information. Inspired by the transformer architecture, we utilize a Multi-channel Adaptive Aggregation Module (MAAM), as depicted in Fig 4.

The output of the dual-channel GCN is utilized to compute cross-attention, which is subsequently input into the residual normalization and feedforward neural network layers. The calculation process can be expressed as follows:

$$\widehat{H^{sem}} = \text{LN}(\text{MHA}(Q^{sem}, K^{syn}, V^{syn}) + \widetilde{H^{sem}}) \tag{12}$$

$$H^{sem} = \text{LN}(\text{FFN}(\widehat{H^{sem}}) + \widehat{H^{sem}}) \tag{13}$$

$$\widehat{H^{syn}} = \text{LN}(\text{MHA}(Q^{syn}, K^{sem}, V^{sem}) + \widetilde{H^{syn}}) \tag{14}$$

$$H^{syn} = \text{LN}(\text{FFN}(\widehat{H^{syn}}) + \widehat{H^{syn}}) \tag{15}$$

where LN(·) denotes layer normalization, FFN(·) is feed-forward neural network, and MHA(·) represents the multi-head attention mechanism. $Q$, $K$, and $V$ are the outputs of $H^{sem}$ and $H^{syn}$ after being processed by the linear layer.

However, during the training process, the model might tend to learn either semantic or syntactic features. Hence, we concatenate $\widehat{H^{sem}}$ and $\widehat{H^{syn}}$ to derive $H^c$:

$$H^c = \text{FFN}(\widehat{H^{sem}} \oplus \widehat{H^{syn}}) \tag{16}$$

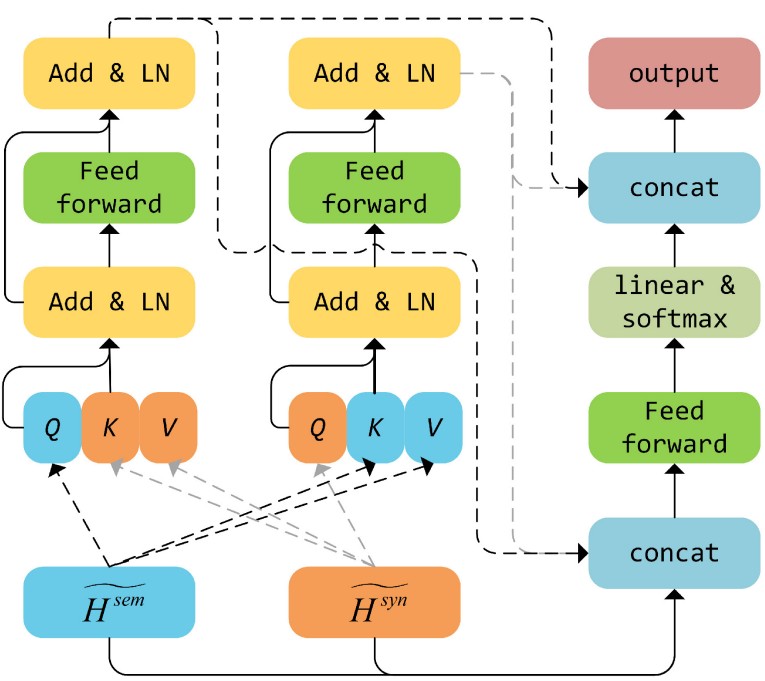

**Fig 4. The overall architecture of MAAM module.**

where $\oplus$ represents the concat operation. To achieve effective integration of $H^{sem}$, $H^{syn}$, and $H^c$, the attention scores associated with these components were weighted and combined utilizing a softmax function. Given the input $H^{stack}=[\,H^{sem}, H^{syn}, H^c]$, the computation for the fusion output is delineated as follows:

$$\alpha_i = \frac{\exp(\sigma(H_i^{stack}W + b))}{\sum_{i=1}^{3}\exp(\sigma(H_i^{stack}W + b))} \tag{17}$$

$$H^{fin} = [\alpha_1 H^{sem} \oplus \alpha_2 H^{syn} \oplus \alpha_3 H^c] \tag{18}$$

where $\sigma$ denotes the activation function, we use ReLU($\cdot$), $W$ and $b$ represent the learnable weight and bias matrices, respectively, $\oplus$ denotes the concat operation. The above approach allows for a complete interaction between the semantic and syntactic features, resulting in the final output, $H^{fin}$.

To obtain the contextual feature for the specific aspect, we employ a zero mask mechanism. This approach preserves the context vector while replacing the aspect word vector with zeros. Consequently, we derive aspect-specific contextual feature $H^{mask}$, $H^{mask} = \{h_1^{fin}, h_2^{fin}, h_3^{fin}, \cdots, 0, \cdots, h_n^{fin}\}$.

$$h_i^{mask} = \begin{cases} 0, & t+1 \le i \le t+m \\ h_i^{fin} & 1 \le i < t+1, t+m < i \le n \end{cases} \tag{19}$$

## Output layer

We construct the final classification representation $H^{out}$ of the model based on the output of the multi-head attention mechanism from the semantic features $H^{att}$ and the output of the

aspect-level masking mechanism $H^{mask}$. Subsequently, we employ a linear layer followed by a softmax function to compute the sentiment probability distribution $y$:

$$H^{out} = [f(H^{att}) \oplus f(H^{mask})] \tag{20}$$

$$y_{(s,a)} = \text{softmax}(W^o H^{out} + b^o) \tag{21}$$

where $f(\cdot)$ represents the average pooling function, $\oplus$ indicates vector concatenation, $W^o$ and $b^o$ respectively denote the learnable weight and bias matrices, and $(s,a)$ represents the sentence aspect pair.

## Multi-strategy auxiliary module

**Affective refinement strategy.** The majority of methods related to the ABSA task that employs sentiment lexicons tends to combine dependency relationships to incorporate sentiment information, often neglecting the importance of sentiment information within the semantic context. In response to this gap, we propose an affective refinement strategy based on the SenticNet 8 [31] sentiment lexicon. Through a systematic traversal of sentences within the corpus and the integration of the SenticNet 8 sentiment lexicon, we develop an affective score vector, denoted as $lex$, for each sentence. $lex = \{score_1, score_2, score_3, \cdots, score_n\}$.

$$lex_i = \begin{cases} score_i, & \text{if } w_i \text{ in senticNet 8} \\ 0, & \text{otherwise} \end{cases} \tag{22}$$

where $score_i$ represents the sentiment score of the word $i$ in the corresponding sentiment lexicon. If word $i$ does not exist in the sentiment lexicon, then $score_i = 0$. To enhance the model's ability to recognize sentiment cues in sentences, we establish a mapping from the output Hfin of the MAAM layer to $H^s$. By employing the Mean Squared Error (MSE) function to minimize the discrepancy between $H^s$ and the sentence sentiment vector $lex$, we derived the sentiment refinement loss, denoted as $Loss_{ar}$, the calculation process is as follows:

$$H^s = H^{out} W^s + b^s \tag{23}$$

$$Loss_{ar} = \text{MSE}(H^s, lex) \tag{24}$$

where $W^s$ and $b^s$ denote the learnable weight and bias matrices, respectively.

**Supervised contrastive learning strategy.** To enhance the model's ability in inter-aspect modeling and distinguishing inter-class relationships, we employ supervised contrastive learning to assist in model training. Specifically, we regard the aspect representations with the same sentiment polarity in a mini-batch as positive examples and those with different aspect representations as negative examples.

$$Loss_{scl} = -\sum_{i \in B} \frac{1}{C(i)} \sum_{y_i = y_c, c \neq i} \frac{\exp(h_i^a \cdot h_c^a / \tau)}{\sum_{b \in B, b \neq i} \exp(h_i^a \cdot h_b^a / \tau)} \tag{25}$$

where $i$ indicates the index value in the mini-batch sample $B$. $C(i)$ represents the number of positive samples of the $i$-th aspect. $h^a$ denotes the aspect term representation in Formula 21. $\tau$ represents the temperature.

## Model training

We utilize the cross-entropy loss function with L2 regularization to train the model and derive the cross-entropy loss $Loss_{sa}$.

$$Loss_{sa} = - \sum_{(s,a) \in D} \sum_{c \in C} \log y_{(s,a)} + \lambda \|\theta\|^2 \tag{26}$$

where $D$ and $C$ denote the training sample pairs and the set of different sentiment categories respectively. $\lambda$ stands for the L2 regularization parameter.

The final loss $Loss_{total}$ is acquired by weighting the model training loss $Loss_{sa}$, the affective refinement loss $Loss_{ar}$, and the supervised contrastive loss $Loss_{scl}$, which can be computed as follows:

$$Loss_{total} = Loss_{sa} + \alpha Loss_{ar} + \beta Loss_{scl} \tag{27}$$

where $\alpha$ represents the weighting parameter of the affective refinement loss $Loss_{ar}$, and $\beta$ is the weighting parameter of the sample supervised contrastive loss $Loss_{scl}$. The goal of model training is to minimize the total loss $Loss_{total}$.

## Experiment

### Datasets

We conduct evaluations using four publicly available English datasets for the ABSA task, namely the Twitter dataset from ACL14 [43], and the Rest14, Rest15, and Rest16 datasets from SemEval 2014 [44], SemEval 2015 [45], and SemEval 2016 [46], respectively. The Rest14, Rest15, and Rest16 datasets contain statements that might comprise one or multiple aspect terms, while the Twitter dataset contains a single aspect term. The sentiment polarities within these datasets are classified into positive, negative, or neutral. Detailed specifications for each dataset are presented in Table 1.

### Implementation details and training parameters

In the experimental design, we utilize the pre-trained 'bert-base-uncased' version of BERT to generate word embeddings. The input construction for BERT follows the Bert-SPC methodology, with the dimension of the hidden layer being 768. Both the SemGCN and SynGCN architectures consist of two layers each. The weighting value $\alpha$ for the affective refinement strategy lies within the interval (0, 1]. The weighting value $\beta$ in the supervised contrastive strategy also falls within the interval (0, 1]. During the model training process, we initialize the parameter weights using a uniform distribution. The batch size is set to 16, the learning rates are 2e-5 and 5e-5 respectively, and the value of the L2 regularization parameter is 1e-5. We utilize accuracy and macro F1 value metrics to evaluate the performance of the model

**Table 1. The scale of each dataset.**

| Dataset | Trainset | | | Testset | | |
|---|---|---|---|---|---|---|
| | Positive | Neutral | Negative | Positive | Neutral | Negative |
| Rest14 | 2164 | 637 | 807 | 728 | 196 | 196 |
| Twitter | 1561 | 3127 | 1560 | 173 | 346 | 173 |
| Rest15 | 912 | 36 | 256 | 326 | 34 | 182 |
| Rest16 | 1240 | 69 | 439 | 469 | 30 | 117 |

in our experiment. Among them, the accuracy is obtained by calculating the proportion of the total number of correctly predicted samples in the model, while the macro F1 value is derived by calculating the metrics of each label and finding their unweighted average. During the experiment, the optimal performance parameters on each dataset are presented in Table 2. The epoch is set at 30. If no better performance metrics emerge for five consecutive rounds during the training process, the training will be prematurely stopped. All the experiments in our study are conducted on the NVIDIA GeForce RTX 3090 GPU.

## Baselines

To assess the efficacy of the SDMAE model, we refer to three types of baselines: 1) Neural Network-based methods, 2) Graph-based methods, and 3) Graph-based with BERT methods. We conduct comparisons with the following baseline models on the four datasets.

- **TD-LSTM**: Tang et al. [16] design two LSTMs to obtain the context information of aspect words and then concatenate them to predict sentiment polarity.
- **AOA**: Huang et al. [18] employ the idea of attention over attention to model the relationship between aspects and the context for predicting sentiment polarity.
- **IAN**: Ma et al. [19] propose an interactive attention network to jointly model aspect word and context representations to predict sentiment polarity.
- **RAM**: Chen et al. [20] utilize multiple attention mechanisms to capture the sentiment features of specific aspect words in long-distance texts, effectively reducing the influence of irrelevant factors.
- **MemNet**: Tang et al. [47] develop a memory network, combining the attention mechanism with the external memory to calculate the importance of context word to the aspect term.
- **BERT**: Song et al. [42] employ BERT's final hidden state to encode and represent the context for sentiment prediction.
- **ASGCN**: Zhang et al. [8] utilize GCN to model syntactic graph based on dependency tree and combined attention mechanisms to predict sentiment.
- **CDT**: Sun et al. [48] exploit BiLSTM to learn the semantics of contextual words and model the syntactic structure information of dependency trees through convolution.
- **BiGCN**: Zhang et al. [23] develop an interactive bidirectional graph convolutional network to integrate word co-occurrence information and syntactic features.
- **kumaGCN**: Chen et al. [49] achieve representations tailored for the ABSA task through the integration of syntactic dependency graphs and latent graphs.
- **R-GAT**: Wang et al. [50] propose a relational graph attention network and an aspect-oriented tree structure that concentrates on aspects by reshaping and pruning dependency trees.

**Table 2. The optimal parameter combination on each dataset.**

| parameter | Rest14 | Twitter | Rest15 | Rest16 |
|---|---|---|---|---|
| *learning rate* | 2e-5 | 5e-5 | 2e-5 | 5e-5 |
| $\sigma$ | 1.0 | 1.0 | 1.0 | 1.0 |
| $\alpha$ | 0.8 | 0 | 0.7 | 0.7 |
| $\beta$ | 0.3 | 0.3 | 0.3 | 0.5 |
| $\tau$ | 0.07 | 0.07 | 0.07 | 0.07 |
| *max_length* | 100 | 100 | 100 | 100 |

- **ACLT**: Zhou et al. [51] develop aspect-centered tree structure to adaptively correlate aspects with opinion words, thereby reducing the distance between the aspect and the corresponding opinion words.
- **DGEDT**: Tang et al. [52] integrate BiGCN and the transformer structure to acquire the architecture of sentiment features from various perspectives. Subsequently, BiAffine was employed for feature fusion.
- **Dual-GCN**: Li et al. [25] introduce a dual-channel GCN that separately models both the semantic and syntactic graphs. The framework captures semantic correlations through the application of orthogonal and differential regularizers.
- **SenticGCN**: Liang et al. [11] incorporate sentiment knowledge from the SenticNet lexicon into syntactic structures for the enhancement of the model's sentiment perception capability.
- **ASHGAT**: Ouyang et al. [53] construct a word-level hypergraph matrix based on syntactic dependencies to enhance the syntactic and semantic connections between the aspect and the contextual words.
- **TCKGCN**: Hao et al. [13] integrate semantic, conceptual, and affective features and proposed a three-channel knowledge fusion model for sentiment analysis, which strengthened the optimization of aspect and context coordination through an interactive attention mechanism.

## Experimental results and analysis

From the experimental outcomes presented in Table 3, it is evident that SDMAE has achieved the optimal performance on the Rest14, Rest15, and Rest16 datasets when compared with

Table 3. The performance results of four datasets on different models are shown below, with data sourced from copied paper authors and code provided by some published papers. The best performance is marked in bold, the second best performance is indicated in italics, and the dash ('-') indicates that the original research report did not include results or research findings.

| Category | Models | Rest14 | | Twitter | | Rest15 | | Rest16 | |
|---|---|---|---|---|---|---|---|---|---|
| | | Acc | F1 | Acc | F1 | Acc | F1 | Acc | F1 |
| Neural Network based | TD-LSTM [16] | 78.00 | 66.73 | 70.80 | 69.00 | 76.39 | 58.70 | 82.16 | 54.21 |
| | AOA [18] | 79.97 | 70.42 | 72.30 | 70.20 | 78.17 | 57.02 | 87.50 | 66.21 |
| | IAN [19] | 79.26 | 70.09 | 72.50 | 70.81 | 78.54 | 52.65 | 84.74 | 55.21 |
| | RAM [20] | 78.48 | 68.54 | 70.09 | 66.48 | 79.98 | 60.57 | 83.88 | 62.14 |
| | MemNet [47] | 79.60 | 69.60 | 71.50 | 69.00 | 77.30 | 58.30 | 85.40 | 66.00 |
| | BERT [42] | 84.11 | 76.68 | 75.52 | 73.23 | 83.48 | 66.18 | 90.10 | 74.16 |
| Graph-based | ASGCN [8] | 80.77 | 72.02 | 72.15 | 70.40 | 79.89 | 61.89 | 88.99 | 67.48 |
| | CDT [48] | 82.30 | 74.02 | 74.66 | 73.66 | – | – | – | – |
| | BiGCN [23] | 81.97 | 73.48 | 74.16 | 73.35 | 81.16 | 64.79 | 88.96 | 70.84 |
| | kumaGCN [49] | 81.43 | 73.64 | 72.45 | 70.77 | 80.69 | 65.99 | 89.39 | 73.19 |
| | R-GAT [50] | 83.30 | 76.08 | 75.57 | 73.82 | 80.83 | 64.17 | 88.92 | 70.89 |
| | ACLT [51] | 85.71 | 78.44 | 75.48 | 74.51 | 82.44 | 72.08 | 92.15 | 78.64 |
| | DGEDT [52] | 83.90 | 75.10 | 74.80 | 73.40 | 82.10 | 65.90 | 90.80 | 73.80 |
| | Dual-GCN [25] | 84.27 | 78.08 | 75.92 | 74.29 | – | – | – | – |
| | SenticGCN [11] | 84.03 | 75.38 | – | – | 82.84 | 67.32 | 90.88 | 75.91 |
| | TCKGCN [13] | – | – | 75.92 | 74.26 | 82.44 | 66.12 | 90.37 | 71.79 |
| Graph-based with BERT | Dual-GCN+BERT | *87.13* | *81.16* | *77.40* | *76.02* | – | – | – | – |
| | DGEDT+BERT [52] | 86.30 | 80.00 | **77.90** | 75.40 | 84.00 | 71.00 | 91.90 | 79.00 |
| | SenticGCN+BERT [11] | 86.92 | 81.03 | – | – | *85.32* | *71.28* | *91.97* | *79.56* |
| | ASHGAT+BERT [53] | 85.49 | 79.23 | – | – | 83.57 | 71.15 | 90.75 | 77.57 |
| | TCKGCN+BERT [13] | – | – | 77.39 | **76.30** | 84.25 | 69.53 | 91.47 | 76.71 |
| | **SDMAE** | **87.59** | **81.94** | 76.88 | 75.37 | **88.19** | **75.78** | **92.86** | **82.76** |

previous research methodologies. Additionally, its performance on the Twitter dataset is comparable to that of previous methods. These findings demonstrate the efficacy of the SDMAE method. After the comparison, the following results are obtained.

Firstly, within the neural network-based methodologies, when compared to the TD-LSTM models, the IAN and AOA models demonstrate significantly superior performance across all datasets. This is because the IAN and AOA methods provide specific aspect information to the model. It has been affirmed that integrating such information is beneficial for the model to retrieve key information within the context. Among the neural network methods, BERT performs the best, suggesting that the semantic features obtained by BERT are more effective than those acquired by other approaches. The graph-based method has enhanced the overall performance, indicating the necessity of incorporating syntactic structure information in the ABSA task. This is because the introduction of syntactic dependency trees infuses syntactic knowledge into the model. By combining syntactic knowledge, the model can learn opinion word information that has a long-distance dependency relationship with aspect terms.

In comparison with the ASGCN, the BiGCN showcases enhanced performance, suggesting that incorporating word co-occurrence information within the corpus is effective. R-GAT trims dependency trees and introduces attention mechanisms into graph neural networks to dynamically adjust node weights within the syntactic graph. In contrast to ASGCN, a remarkable improvement in performance has been observed in the Twitter and Rest14 datasets, thereby validating that pruning specific aspects is advantageous. ACLT reduces the distance between aspects and their corresponding opinion words by learning aspect-centered tree structures. Compared to standard dependency trees, it can adaptively associate aspects and opinion words during model training. The performance across all five datasets has been improved. Notably, the accuracy of the Rest16 dataset has reached 92.15%, indicating that minimizing the distance between aspects and corresponding opinion words enables the model to better recognize the sentiment polarity in the ABSA task.

SenticGCN and TCKGCN integrate external knowledge with semantic information or syntactic dependency trees. In comparison to the ASGCN model, the SenticGCN model demonstrated an average enhancement of 2.70% in accuracy and 5.74% in macro F1 score, respectively, on the Rest14, Rest15, and Rest16 datasets. TCKGCN exhibited an average improvement of 2.57% in accuracy and 4.13% in macro F1 values on the datasets excluding Rest14. It has been validated that external knowledge can direct models to learn sentiment and common-sense information, assist models in sentiment classification, and notably enhance the performance of the ABSA task. At the same time, in the graph-based methods, those using BERT also perform better, indicating the superiority of BERT in semantic encoding.

## Ablation experiment

**The impact of various components in SDMAE.** To further investigate the influence of each component of the model on performance, an ablation experiment was conducted: 1) w/o AOSD denotes deleting the syntactic denoising graph and utilizing the original syntactic dependency graph. 2) w/o $Loss_{ar}$, w/o $Loss_{scl}$, and w/o $Loss_{ar}$ & $Loss_{scl}$ respectively represent removing only the sentiment refinement loss, removing only the sample supervised contrast loss, and simultaneously removing both and computing only the classification loss of the model. The results of the ablation experiment are presented in Table 4.

On the one hand, in the case of w/o AOSD where the original dependency tree was utilized to generate graphs, a significant performance decline was observed on the Rest14, Rest15, and

**Table 4. Experimental results of ablation of various components in SDMAE, the best performance is marked in bold.**

| Models | Rest14 | | Twitter | | Rest15 | | Rest16 | |
|---|---|---|---|---|---|---|---|---|
| | Acc | F1 | Acc | F1 | Acc | F1 | Acc | F1 |
| SDMAE | **87.59** | **81.94** | **76.88** | **75.37** | **88.19** | **75.78** | **92.86** | **82.76** |
| w/o AOSD | 86.61 | 80.78 | 76.27 | 75.19 | 86.16 | 73.95 | 91.23 | 77.56 |
| w/o $Loss_{ar}$ | 86.79 | 81.05 | 76.88 | 75.03 | 86.90 | 73.10 | 91.56 | 77.77 |
| w/o $Loss_{scl}$ | 87.05 | 81.33 | 75.72 | 74.67 | 86.53 | 75.11 | 92.69 | 80.76 |
| w/o $Loss_{ar}$ & $Loss_{scl}$ | 87.23 | 81.52 | 75.72 | 74.67 | 86.35 | 76.83 | 91.23 | 79.24 |

Rest16 datasets. However, the performance on the Twitter dataset remained nearly equivalent to that when AOSD was employed. This suggests that within the Twitter dataset, the proportion of noisy information in the syntactic structure is relatively low. On the other hand, we individually applied an affective refinement strategy and sample supervised contrastive learning for ablation. It can be noted that these two strategies have different contributions in various datasets. When both strategies are simultaneously removed, a relatively substantial performance degradation occurs compared to the complete method. Evidently, removing any module will impact the performance of the model and lead to varying degrees of decline.

**The different fusion ways of semantic and syntactic information.** To demonstrate the efficacy of the MAAM layer in information aggregation, we conduct ablation experiments by utilizing three common information aggregation methods in place of MAAM. w/o MAAM indicates the removal of the MAAM layer and its substitution with three common information aggregation methods namely "sum", "concat" and "gate". As illustrated in Fig 5, in the "gate" aggregation, the LeakyReLU activation function is utilized to compute the gate control output.

It can be clearly observed from Table 5 and Fig 6, in the Rest14, Twitter, and Rest15 datasets, there were varying degrees of performance deterioration in terms of accuracy and macro F1 values upon the removal of the MAAM layer. Compared to the MAAM layer, these aggregation methods were incapable of fully aggregating semantic and syntactic features. In the Rest16 dataset, although the "concat" method achieved higher accuracy, there was a considerable decrease in macro F1 values, and the accuracy with the MAAM layer was comparable to it.

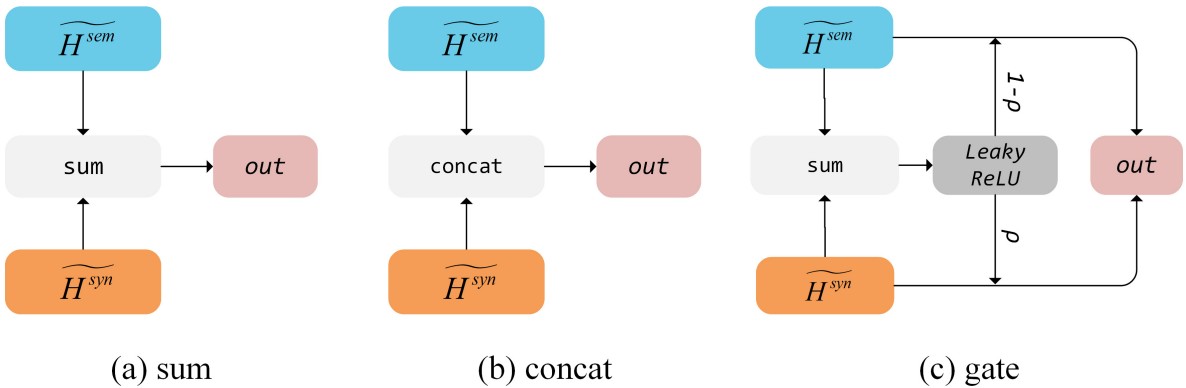

(a) sum  (b) concat  (c) gate

**Fig 5. Different aggregation methods of semantic and syntactic information.**

**Table 5. Comparison of effectiveness between MAAM module and common aggregation methods, the best performance is marked in bold.**

| Models | Rest14 | | Twitter | | Rest15 | | Rest16 | |
|---|---|---|---|---|---|---|---|---|
| | Acc | F1 | Acc | F1 | Acc | F1 | Acc | F1 |
| **SDMAE** | **87.59** | **81.94** | **76.88** | **75.37** | **88.19** | **75.78** | 92.86 | **82.76** |
| w/o MAAM (with sum) | 86.43 | 79.97 | 75.43 | 74.29 | 87.45 | 75.68 | 90.75 | 76.45 |
| w/o MAAM (with concat) | 85.09 | 78.90 | 74.42 | 72.77 | 84.69 | 73.85 | **93.34** | 77.56 |
| w/o MAAM (with gate) | 86.79 | 81.05 | 76.59 | 75.03 | 85.79 | 73.90 | 91.23 | 77.38 |

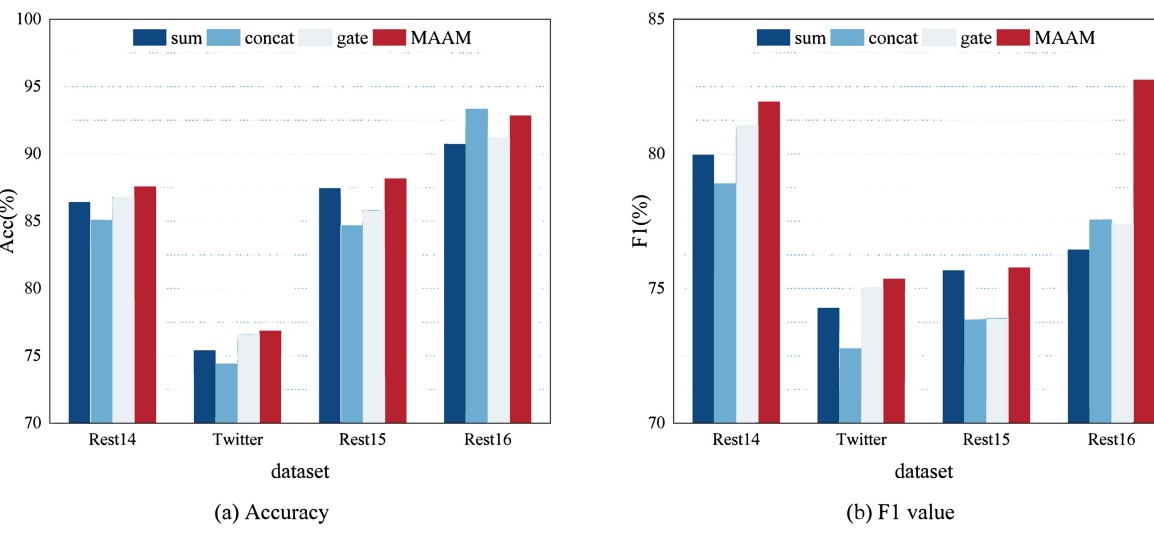

(a) Accuracy                    (b) F1 value

**Fig 6. The variation between accuracy and F1 value under different aggregation methods.**

**The impact of MHA headcount.** In addition, we also investigated the influence of the number of heads in the multi-head attention on the experimental performance. Visualization results of the ablation experiment on the number of MHA heads are shown in Fig 7 as follows. We increased the number of attention heads from 2 to 12 and conducted experiments on four publicly available datasets. As indicated by the results, when the number of attention heads is 6, the model we proposed attains the optimal accuracy on all datasets. As the number of attention heads increases, the performance of the model commences to deteriorate. Similarly for the macro F1 value, when the number of attention heads is 6, Twitter, Rest15, and Rest16 achieve the optimal macro F1 value; Rest14 performs optimally when the number of attention heads is 8. It can be discerned that when the number of attention heads is 6, the model can achieve superior performance. Once it exceeds or is lower than this threshold, the performance begins to decline. The reason is that as the number of attention heads increases, the learning ability of the entire model gradually ascends and reaches its peak when the number of attention heads equals 6. Once it exceeds the threshold, the features of the nodes become overly cumbersome. Assigning a higher weight to each head will cause the model to lose the ability to select important nodes.

**The visualization matrix of the AOSD algorithm.** To visually present the AOSD algorithm, we employ sentence examples for visualization. Fig 8 depicts the original adjacency matrix A and the adjacency matrix M after denoising by the AOSD algorithm for the sentence "the charging speed of this phone is extremely fast and stable", where the aspect term is the

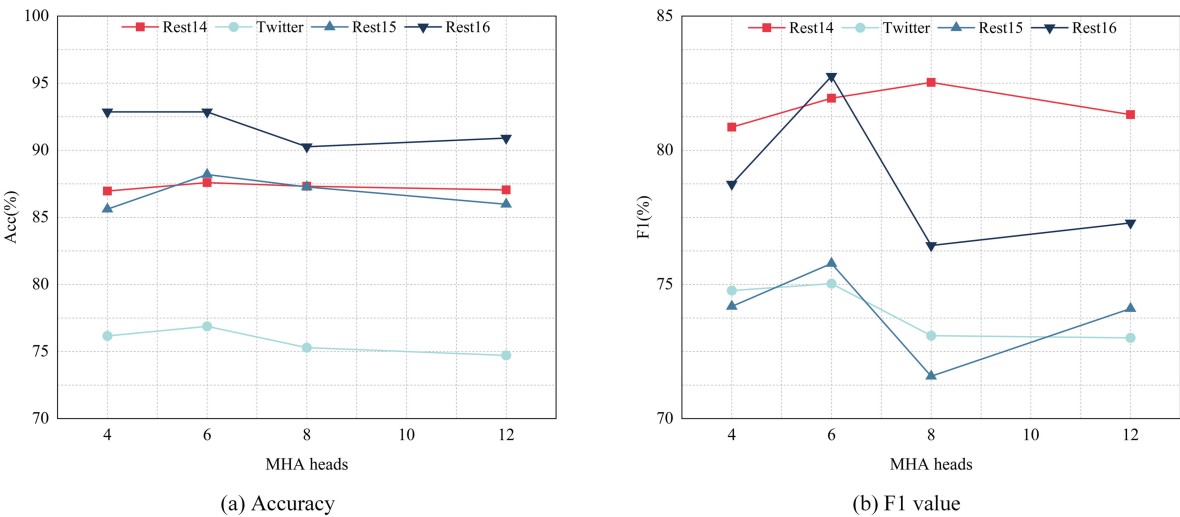

**Fig 7. The variation between accuracy and F1 value under different numbers of MHA heads.**

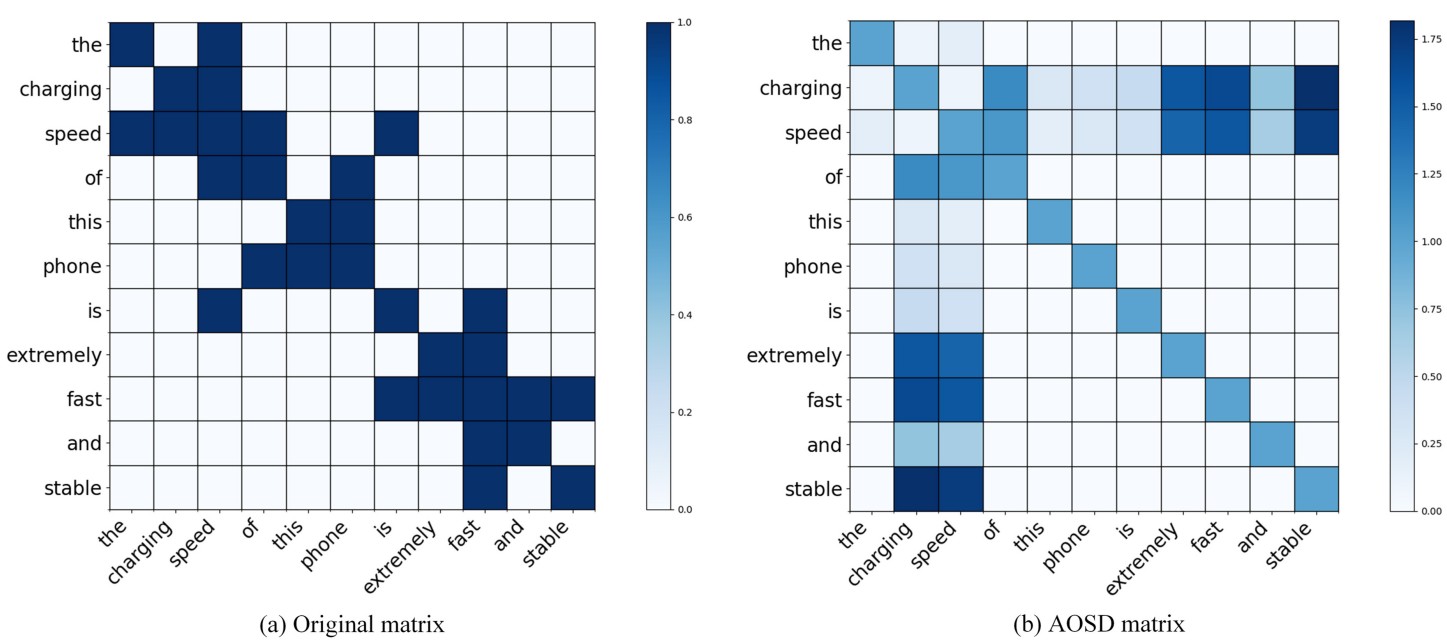

**Fig 8. The visualization matrix of original syntactic and after AOSD of sentence "the charging speed of this phone is extremely fast and stable".**

given "charging speed". It can be observed that in the pruned syntactic matrix following the AOSD denoising algorithm, the degree of attention of the aspect term "charging speed" to the opinion words "extremely", "fast", and "stable" has significantly increased. In comparison with the original syntactic adjacency matrix, it is more conducive for the model to capture the relationship between the opinion words and the aspect term, thereby facilitating the making of the correct sentiment decision. To better analyze the effective help brought by AOSD algorithm to SDMAE, as shown in Table 6, we did a case study of the original matrix and AOSD matrix for two text examples.

**Table 6. Results of case study on the original matrix and AOSD matrix, with the aspect words underlined.**

| Example | w/o AOSD(original) | AOSD |
|---|---|---|
| The <u>dinner</u> was **ok**, **nothing** i would have again. | dinner positive ✗ | dinner negative ✓ |
| The <u>pizza</u> is very **good**, so is the atmosphere. | pizza positive ✓atmosphere positive ✓ | pizza positive ✓atmosphere positive ✓ |
| The <u>charging speed</u> of this phone is **extremely fast** and **stable** | charging speed negative ✗ | charging speed positive ✓ |

## Discussion

The goal of this study was to tackle two critical challenges in ABSA, namely syntactic noise interference and insufficient integration of semantic-syntactic information, while validating the effectiveness of the proposed SDMAE method. Through the collaborative design of multiple components, SDMAE addresses these research goals in a targeted manner, as elaborated below.

### Syntactic noise

Online review data often contains noise due to non - standard syntactic structures, which interferes with the model's ability to capture sentiment - related relationships. The AOSD algorithm trims the original syntactic dependency tree by filtering part-of-speech (prioritizing VERB, ADV, ADJ, etc., which are strongly associated with sentiment) and applying non-linear position weighting relative to aspect terms. Ablation experiments show that when AOSD is removed, the accuracy on the Rest14, Rest15, and Rest16 datasets decreases by 0.98%, 2.03%, and 1.63% respectively, and the macro-F1 values drop by 1.16%, 1.83%, and 5.2% respectively. Take the sentence "The charging speed of this phone is extremely fast and stable" as an example. AOSD enhances the syntactic association weights between "charging speed" and "fast"/"stable", enabling the model to more accurately focus on key components related to sentiment, directly addressing the issue of "syntactic noise interference".

### Semantic-syntactic fusion

The inefficient integration of semantic and syntactic information can limit the model's understanding of sentiment relationships. The MAAM Module in SDMAE achieves dynamic interaction between semantic feature and syntactic features through a multi-head attention mechanism. Compared with traditional fusion methods, MAAM shows more stable performance across datasets. On the Rest14 dataset, the macro-F1 of MAAM is 1.97% higher than that of the "sum" method; on the Rest15 dataset, the accuracy (Acc) is 3.5% higher than that of the "concat" method. This adaptive fusion mechanism allows the model to flexibly balance the contributions of semantics and syntax, effectively overcoming the problem of "insufficient integration of semantic-syntactic information" and verifying the rationality of the SDMAE design.

### Threats to validity and limitations

Despite the promising performance of SDMAE, several factors related to the design, data, and methodology may affect the validity and generalizability of the conclusions. These threats are outlined as follows:

**Construct Validity.** The sentiment annotation of the datasets relies on manual judgment, which introduces subjectivity. Ambiguous sentiment expressions (e.g., "not bad") may lead to

annotation inconsistencies due to differences in annotators' interpretations. Furthermore, the part-of-speech filtering strategy in the AOSD module (prioritizing VERB, ADV, and ADJ) is based on general sentiment heuristics. Such heuristics may not be universally applicable in all domains, potentially limiting the accuracy of sentiment structure extraction.

**Internal Validity.** The performance of SDMAE depends on hyperparameters such as the number of attention heads and the weighting of multi-strategy losses, which are tuned on benchmark datasets. Although these configurations lead to strong empirical results, they also pose a risk of overfitting to the experimental setup. Changes in hyperparameter settings (e.g., reducing the number of attention heads) may result in performance variability, thereby threatening the internal consistency of the conclusions.

**External Validity.** The experiments are conducted exclusively on English-language online review datasets. The model's generalization ability to other languages or to different types of text (e.g., formal writing, domain-specific documents) has not been evaluated. This lack of validation across diverse data sources limits the external applicability of the findings.

**Limitations of the SDMAE Model.** Although SDMAE introduces several innovations, it still exhibits certain limitations: (1) *Insufficient Coverage of Sentiment Knowledge*: The sentiment refinement relies on SenticNet 8, which lacks complete coverage of domain-specific terminology, leading to suboptimal sentiment vector representations. (2) *Inadequate Handling of Complex Sentences*: The MAAM module struggles to distinguish sentiment associations in sentences with multiple conflicting aspects (e.g., "The food is good but the service is bad"). The absence of an aspect-specific hierarchical attention mechanism hinders accurate aspect-opinion alignment. (3) *Unverified Domain Adaptability*: While SDMAE performs well on general review datasets, its effectiveness in professional or domain-specific contexts remains untested. The use of a general-domain BERT encoder may fail to capture fine-grained sentiment cues in specialized domains.

## Conclusions and future work

In this paper, we propose the SDMAE method to tackle the issues of syntactic noise and inefficient fusion of semantic and syntactic information in the ABSA task. We address these challenges by integrating the aspect-oriented syntactic denoising algorithm and a multi-strategy auxiliary approach. Specifically, AOSD effectively prunes the dependency trees to reduce noise by incorporating the position of aspect words, the part-of-speech of the context, and the feature information of contextual distance. The Multi-channel Adaptive Aggregation Module combines the weights of semantic and syntactic features in the model, facilitating the effective integration of these features through attention mechanisms. Furthermore, the affective refinement strategy and supervised contrastive learning strategy are employed to enhance the model's ability in sentiment recognition and inter-class discrimination, respectively. Experiments conducted on four benchmark datasets demonstrate that SDMAE outperforms previous methods, confirming its effectiveness.

We believe that although our model fully leverages the emotional knowledge of words and contextual distance information, relying solely on the SenticNet emotional lexicon to provide sufficient emotional knowledge for the ABSA (Aspect-Based Sentiment Analysis) task remains inadequate. Further exploration and research on the acquisition and utilization of emotional knowledge are required. In future work, we aim to enhance the model's emotional recognition capabilities by considering emotional interaction effects and leveraging the prompt-learning abilities of Large Language Models, integrating more relevant information from various dimensions.

## Author contributions

**Conceptualization:** Lu Liu, Xiaojin Gao.

**Data curation:** Yunhai Zhu.

**Formal analysis:** Lu Liu, Chuanxu Yue.

**Funding acquisition:** Xiaojin Gao, Yunhai Zhu.

**Investigation:** Da Li.

**Methodology:** Lu Liu.

**Project administration:** Xiaojin Gao, Yunhai Zhu.

**Resources:** Xiaojin Gao.

**Software:** Lu Liu.

**Supervision:** Da Li.

**Validation:** Da Li, Yunhai Zhu.

**Visualization:** Chuanxu Yue.

**Writing – original draft:** Lu Liu.

**Writing – review & editing:** Lu Liu, Chuanxu Yue, Xiaojin Gao, Yunhai Zhu.

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
