## [Decision Letter · Decision Letter 0]

30 Apr 2025

PONE-D-25-17433Syntactic denoising and multi-strategy auxiliary enhancement for Aspect-based Sentiment AnalysisPLOS ONE

Dear Dr. Zhu,

Thank you for submitting your manuscript to PLOS ONE. After careful consideration, we feel that it has merit but does not fully meet PLOS ONE’s publication criteria as it currently stands. Therefore, we invite you to submit a revised version of the manuscript that addresses the points raised during the review process.

 As you can see, the reviewers have a concern about missing major sections. Kindly consider all the attached comments attached with this email including those raised by the academic editor.

We look forward to receiving your revised manuscript.

Kind regards,

Issa Atoum

Academic Editor

PLOS ONE

“The Key R&D Program (Science and Technology Cooperation) of Shandong Province (2024KJHZ030).

The Innovation Pilot Project for the Integration of Science, Education, and Industry (2024GH12).

The Innovation Capability Enhancement Project for Science and Technology oriented Small and Medium sized Enterprises in Shandong Province (2024TSGC0903).”

“This study is funded by the Key R&D Program (Science and Technology Cooperation)

of Shandong Province (2024KJHZ030), the Innovation Pilot Project for the Integration

of Science, Education, and Industry (2024GH12), and the Innovation Capability

Enhancement Project for Science and Technology oriented Small and Medium sized

Enterprises in Shandong Province (2024TSGC0903).”

“The Key R&D Program (Science and Technology Cooperation) of Shandong Province (2024KJHZ030).

The Innovation Pilot Project for the Integration of Science, Education, and Industry (2024GH12).

The Innovation Capability Enhancement Project for Science and Technology oriented Small and Medium sized Enterprises in Shandong Province (2024TSGC0903).”

Additional Editor Comments:

1- Please have a dedicated section for discussion that is aligned with the research objectives.

2- Add a separate section or subsections for threats to validity , limitations and implication of the study.

Reviewers' comments:

Reviewer's Responses to Questions

**Comments to the Author**

1. Is the manuscript technically sound, and do the data support the conclusions?

Reviewer #1: Yes

2. Has the statistical analysis been performed appropriately and rigorously? 

Reviewer #1: Yes

3. Have the authors made all data underlying the findings in their manuscript fully available?

Reviewer #1: Yes

4. Is the manuscript presented in an intelligible fashion and written in standard English?

Reviewer #1: Yes

5. Review Comments to the Author

Reviewer #1: The paper presents a novel algorithm and it has outperformed the existing models. The work is state of the art but I recommend a few changes for improving the paper.

Here are the recommendations:

1. The error analysis is not presented in the paper. It is recommended to present it and describe each case.

2. It is not clear if hyperparameter tuning is performed. The authors have presented only one set of hyper parameters of each dataset. Having the same parameters for all the models shows that the hyper parameter tuning is not performed.

3. It is recommended to present the limitations of the work.

4. In the figure only AOSD matrix is presented. What about others? It is recommended to present and compare with others as well.

5. At the end of the related work section mentioning the uniqueness of the work is important. So it will be easier to follow and express the uniqueness properly.

6. The motivation of the work should be mentioned in the introduction and how they have arrived at the approach.

7. The conclusion section is not explicitly mentioned. THough the authors have presented the future work and results. It is suggested to explicitly mention the section.

6. PLOS authors have the option to publish the peer review history of their article (what does this mean?). If published, this will include your full peer review and any attached files.

Reviewer #1: **Yes: **Chandrakanth Puligundla

---

## [Author Response · Author response to Decision Letter 1]

11 Jun 2025

Q1: The error analysis is not presented in the paper. It is recommended to present it and describe each case.

Response1:Thanks for your kind suggestion. We have added a specific case error analysis of the proposed model after the ablation experiment. Combining with the visual diagram of the AOSD algorithm can provide a more intuitive understanding of the errors.

Q2: It is not clear if hyperparameter tuning is performed. The authors have presented only one set of hyper parameters of each dataset. Having the same parameters for all the models shows that the hyper parameter tuning is not performed.

Response2: Thanks for your kind suggestion. Our experimental results are all optimal outcomes obtained through hyperparameter tuning. We provide the optimal parameter values of different models, and for the tuning process, we offer a detailed ablation experiment analysis report of the model, including the ablation experiment on the number of attention heads. The hyperparameters of the algorithm in the training process are not the key innovative part of the model.

Q3: It is recommended to present the limitations of the work.

Response3: Thanks for your kind suggestion. Thanks for your kind suggestion. We noticed that the article lacked a description of the model's limitations, and we have added an explanation of the model's limitations in the summary section of the article.

Q4: In the figure only AOSD matrix is presented. What about others? It is recommended to present and compare with others as well.

Response4: Thanks for your kind suggestion. In the ablation experiment of the AOSD algorithm, the original matrix A is the visual diagram of other models, and the matrix M is the AOSD algorithm matrix.

Q5: At the end of the related work section mentioning the uniqueness of the work is important. So it will be easier to follow and express the uniqueness properly.

Response5: Thank you for your suggestion. We have mentioned the uniqueness of this work at the end of the related work section.

Q6: The motivation of the work should be mentioned in the introduction and how they have arrived at the approach.

Response6: Thank you for your valuable suggestion. The motivation of this work has already been mentioned. It has been highlighted in red in the revised version of the introduction.

Q7: The conclusion section is not explicitly mentioned. THough the authors have presented the future work and results. It is suggested to explicitly mention the section.

Response7: Thank you for your valuable suggestion. We have proposed future work and outcomes, and clearly suggested the direction of future work.

---

## [Decision Letter · Decision Letter 1]

24 Jun 2025

PONE-D-25-17433R1Syntactic denoising and multi-strategy auxiliary enhancement for Aspect-based Sentiment AnalysisPLOS ONE

Dear Dr. Zhu,

Thank you for submitting your manuscript to PLOS ONE. After careful consideration, we feel that it has merit but does not fully meet PLOS ONE’s publication criteria as it currently stands. Therefore, we invite you to submit a revised version of the manuscript that addresses the points raised during the review process.

While the authors have addressed the reviewers’ comments, they have not responded to several key points raised by the Academic Editor. Please ensure that all editorial comments are fully addressed. Furthermore, the manuscript does not currently meet PLOS ONE’s data availability requirements: the provided URLs do not lead to the underlying dataset or source code. Please update the links to ensure direct and unrestricted access to both the data and code, as per the journal's policy. 

Please address all comments constructively and revise the manuscript accordingly. Ensure all figures and tables are embedded within the main text. If any suggestion cannot be fulfilled, provide a clear and reasonable justification. Responses should follow the journal’s format and be submitted in a separate response document, with edits highlighted in yellow in the revised manuscript. 

=== From previous decision. ===

“The Key R&D Program (Science and Technology Cooperation) of Shandong Province (2024KJHZ030).

The Innovation Pilot Project for the Integration of Science, Education, and Industry (2024GH12).

The Innovation Capability Enhancement Project for Science and Technology oriented Small and Medium sized Enterprises in Shandong Province (2024TSGC0903).”

“This study is funded by the Key R&D Program (Science and Technology Cooperation)

of Shandong Province (2024KJHZ030), the Innovation Pilot Project for the Integration

of Science, Education, and Industry (2024GH12), and the Innovation Capability

Enhancement Project for Science and Technology oriented Small and Medium sized

Enterprises in Shandong Province (2024TSGC0903).”

“The Key R&D Program (Science and Technology Cooperation) of Shandong Province (2024KJHZ030).

The Innovation Pilot Project for the Integration of Science, Education, and Industry (2024GH12).

The Innovation Capability Enhancement Project for Science and Technology oriented Small and Medium sized Enterprises in Shandong Province (2024TSGC0903).”

Additional Editor Comments:

1- Please have a dedicated section for discussion that is aligned with the research objectives.

2- Add a separate section or subsections for threats to validity , limitations and implication of the study.

==

We look forward to receiving your revised manuscript.

Kind regards,

Issa Atoum

Academic Editor

PLOS ONE

Reviewers' comments:

Reviewer's Responses to Questions

**Comments to the Author**

1. If the authors have adequately addressed your comments raised in a previous round of review and you feel that this manuscript is now acceptable for publication, you may indicate that here to bypass the “Comments to the Author” section, enter your conflict of interest statement in the “Confidential to Editor” section, and submit your "Accept" recommendation.

Reviewer #1: All comments have been addressed

2. Is the manuscript technically sound, and do the data support the conclusions?

Reviewer #1: Yes

3. Has the statistical analysis been performed appropriately and rigorously? 

Reviewer #1: Yes

4. Have the authors made all data underlying the findings in their manuscript fully available?

Reviewer #1: Yes

5. Is the manuscript presented in an intelligible fashion and written in standard English?

Reviewer #1: Yes

6. Review Comments to the Author

Reviewer #1: (No Response)

7. PLOS authors have the option to publish the peer review history of their article (what does this mean?). If published, this will include your full peer review and any attached files.

Reviewer #1: **Yes: **Chandrakanth Puligundla

---

## [Author Response · Author response to Decision Letter 2]

7 Jul 2025

Response Letter

Dear editors and reviewers,

We are very grateful for your constructive comments and suggestions for our manuscript entitled “Syntactic denoising and multi-strategy auxiliary enhancement for Aspect-based Sentiment Analysis” (ID: “PONE-D-25-17433”). Your comments are very valuable and helpful for improving our manuscript. In the following, the responses to all the comments are provided one by one.

We have tried our best to make all the revisions clear, and we hope that the revised manuscript can satisfy the requirements for publication.

Sincerely,

Corresponding author.

Yunhai Zhu

Response to the comments of Editor

Editor Comment 1:Please have a dedicated section for discussion that is aligned with the research objectives.

Response :We appreciate this suggestion. In the revised manuscript, we have created a standalone section titled "Discussion" that explicitly aligns with the research objectives. This section highlights how the proposed SDMAE method addresses the two primary challenges identified in the study—syntactic noise interference and inefficient semantic-syntactic integration. We further elaborate on the performance impact of each component (AOSD and MAAM) through empirical evidence, ensuring that the discussion is directly connected to the research goals.

Editor Comment 2:Add a separate section or subsections for threats to validity, limitations, and implication of the study.

Response:Thank you for pointing this out. In response, we have reorganized and expanded the original text into a dedicated section titled “Threats to Validity and Limitations”, which now includes clearly defined subsections addressing:

Construct Validity: discussing possible annotation bias and the generalization of syntactic rules.

Internal Validity: detailing hyperparameter tuning and overfitting risks.

External Validity: clarifying the limitations in generalizing beyond English review data.

Limitations of SDMAE: covering sentiment knowledge coverage, complex sentence structure handling, and domain adaptability.

Response to the comments of Reviewer 1

Q1: The error analysis is not presented in the paper. It is recommended to present it and describe each case.

Response1:Thanks for your kind suggestion. We have added a specific case error analysis of the proposed model after the ablation experiment. Combining with the visual diagram of the AOSD algorithm can provide a more intuitive understanding of the errors.

Q2: It is not clear if hyperparameter tuning is performed. The authors have presented only one set of hyper parameters of each dataset. Having the same parameters for all the models shows that the hyper parameter tuning is not performed.

Response2: Thanks for your kind suggestion. Our experimental results are all optimal outcomes obtained through hyperparameter tuning. We provide the optimal parameter values of different models, and for the tuning process, we offer a detailed ablation experiment analysis report of the model, including the ablation experiment on the number of attention heads. The hyperparameters of the algorithm in the training process are not the key innovative part of the model.

Q3: It is recommended to present the limitations of the work.

Response3: Thanks for your kind suggestion. Thanks for your kind suggestion. We noticed that the article lacked a description of the model's limitations, and we have added an explanation of the model's limitations in the summary section of the article.

Q4: In the figure only AOSD matrix is presented. What about others? It is recommended to present and compare with others as well.

Response4: Thanks for your kind suggestion. In the ablation experiment of the AOSD algorithm, the original matrix A is the visual diagram of other models, and the matrix M is the AOSD algorithm matrix.

Q5: At the end of the related work section mentioning the uniqueness of the work is important. So it will be easier to follow and express the uniqueness properly.

Response5: Thank you for your suggestion. We have mentioned the uniqueness of this work at the end of the related work section.

Q6: The motivation of the work should be mentioned in the introduction and how they have arrived at the approach.

Response6: Thank you for your valuable suggestion. The motivation of this work has already been mentioned. It has been highlighted in red in the revised version of the introduction.

Q7: The conclusion section is not explicitly mentioned. THough the authors have presented the future work and results. It is suggested to explicitly mention the section.

Response7: Thank you for your valuable suggestion. We have proposed future work and outcomes, and clearly suggested the direction of future work.

---

## [Editor Report · Decision Letter 2]

10 Jul 2025

Syntactic denoising and multi-strategy auxiliary enhancement for Aspect-based Sentiment Analysis

PONE-D-25-17433R2

Dear Dr. Zhu,

We’re pleased to inform you that your manuscript has been judged scientifically suitable for publication and will be formally accepted for publication once it meets all outstanding technical requirements.

Kind regards,

Issa Atoum

Academic Editor

PLOS ONE

Additional Editor Comments (optional):

Please adjust the alignment of the lines to ensure they are justified on both sides, and embed the script code link directly within the manuscript.
---

## [Editor Report · Acceptance letter]

PONE-D-25-17433R2

PLOS ONE

Dear Dr. Zhu,

I'm pleased to inform you that your manuscript has been deemed suitable for publication in PLOS ONE. Congratulations! Your manuscript is now being handed over to our production team.

Kind regards,

on behalf of

Dr. Issa Atoum

Academic Editor

PLOS ONE